# A Candidate Gene Association Study for Economically Important Traits in Czech Dairy Goat Breeds

**DOI:** 10.3390/ani11061796

**Published:** 2021-06-16

**Authors:** Michaela Brzáková, Jana Rychtářová, Jindřich Čítek, Zuzana Sztankóová

**Affiliations:** 1Department of Genetics and Breeding of Farm Animals, Institute of Animal Science, 104 00 Prague, Czech Republic; sztankoova.zuzana@vuzv.cz; 2Department of Biology of Reproduction, Institute of Animal Science, 104 00 Prague, Czech Republic; rychtarova.jana@vuzv.cz; 3Department of Genetics and Agricultural Biotechnologies, Faculty of Agriculture, University of South Bohemia, 370 05 Ceske Budejovice, Czech Republic; citek@zf.jcu.cz; 4Department of Infectious Diseases and Preventive Medicine, Veterinary Research Institute, 621 00 Brno, Czech Republic

**Keywords:** candidate genes, milk performance, somatic cell count, goat, *ACACA*, *BTN1A1*, *LPL*, *SCD*

## Abstract

**Simple Summary:**

The milk production traits of goats are economically important. In the Czech Republic, goat milk is processed directly on farms and distributed as cheese. Although goat breeding is not a main focus of animal production in the Czech Republic, it is essential for the agricultural sector. A group of 14 SNPs (single-nucleotide polymorphisms) within four candidate genes (*ACACA*, *BTN1A1*, *LPL*, and *SCD*) were analysed in two Czech dairy goat breeds, White Shorthaired (WSH) goats and Brown Shorthaired (BSH) goats. The SNPs were significantly associated with milk-production traits (daily milk yield, protein, and fat percentage) and somatic cell count. This information may be useful for marker-assisted selection or related techniques to increase the accuracy of selection.

**Abstract:**

Milk production is influenced by many factors, including genetic and environmental factors and their interactions. Animal health, especially udder health, is usually evaluated by the number of somatic cells. The present study described the effect of polymorphisms in the *ACACA*, *BTN1A1*, *LPL*, and *SCD* genes on the daily milk yield, fat, and protein percentages and somatic cell count. In this study, 590 White Shorthaired (WSH) and Brown Shorthaired (BSH) goats were included. SNP genotyping was performed by PCR-RFLP and multiplex PCR followed by SNaPshot minisequencing analysis. The linear mixed model with repeated measurement was used to identify the genetic associations between the studied genes/SNPs and chosen traits. All selected genes were polymorphic in the tested goat populations and showed significant associations with milk traits. Only *BTN1A1* (SNP g.599 A > G) showed a significant association with the somatic cell score. After Bonferroni correction, a significant effect of *LPL* g.300G > A on daily milk yield and fat percentage, *LPL* g.185G > T on protein percentage, and *LPL* G50C, *SCD* EX3_15G > A, and *SCD* EX3_68A > G on fat percentage was found. The importance of environmental factors, such as the herd-year effect, month of milking, and lactation order on all milk performance indicators was confirmed.

## 1. Introduction

Milk and dairy products are essential for human nutrition. Goat milk is rich in minerals, vitamins, and bioactive components, is easily digestible, and contains fewer allergic proteins than cow milk. These characteristics also suggest the possible use of goat milk for therapeutic purposes [1,2]. In Europe, the small ruminant milk industry is not widespread because of the low number of animals and insufficient milk volume in goats compared to cows. However, the number of goats used for milk production is growing due to expanding demand. In many countries, somatic cell count (SCC), as an indicator of milk quality, plays a role in the milk industry [3,4,5]. A high SCC also negatively affects some flavour characteristics of cheese and ice cream due to a more intense caprine flavour [5,6,7].

Many studies have investigated the relationship between genetic background and milk performance [8,9,10,11,12]. Like cows, the heritabilities of milk yield in goats have been found to be low to moderate. E.g., a study in New Zealand (64,604 lactation records from 23,583 does distributed in 21 flocks) estimates heritabilities of 0.25 for milk yield, 0.24 for fat yield, 0.24 for protein yield, and 0.21 for SCS, suggesting the presence of useful heritable variation [8]. Multiple trait selections for these traits could improve the milk revenue produced from successive generations of New Zealand dairy goats. To a similar extent, a study in Germany estimated heritabilities of 0.15–0.31 for milk yield, 0.21–0.34 for fat content, 0.26–0.50 for protein content, and 0.10–0.17 for the persistence of milk yield [9]. The phenotypic and additive genetic correlations between the milk yield persistence and milk yield in kg were highly positive (0.52–0.72); similarly, the correlations between the protein and fat content were 0.45–0.55. The phenotypic correlations between the fat and protein content and the milk yield were negative, −0.13 to −0.26 and −0.21 to −0.36, respectively (n = 16,579 goats, 42,973 lactations). Others reported milk yield heritabilities of 0.10 to 0.29 (n = 529 goats, 15,509 milk yield test-day records) and a direct heritability for protein percentage of 0.441 (518 phenotypic records from the progeny of 48 sires and 131 dams) [10,11]. The genetic correlation between milk production and SCS can range broadly from −0.16 to 0.43, with a large standard error [12]. Non-genetic factors have been analysed as well. In Alpine goats in Croatia, the herd explained 24% of the variability in daily milk yield, 12% of that in fat content, and 9% of that in protein content; values for the herd test day were 17%, 29% and 30%, and those for the permanent environment were 16%, 3% and 5%, respectively [13].

The effect of major genes is the main research focus, similar to studies in other milking species. The consideration of the αS1-casein genotype may improve the model’s efficiency, translating into more accurate genetic parameters and breeding values [14]. In another study, 48 SNPs in αS1, αS2, β, and κ casein genes were included in the evaluation of genetic parameters [15]. Including genetic effects and relationships among these heritable biomarkers may improve the model efficiency, genetic parameters, and breeding values for milk yield and composition; this inclusion could also help optimise selection practices and profitability for components where technological application may be especially relevant for the cheese-making dairy sector [15]. In addition to the study of candidate loci, the genomic approach in the genome-wide association study has also been applied for the detection of genetic regions of interest [16]. They found a total of 43 genome-wide significant SNPs for lactation yields of milk (MY), fat (FY), protein (PY), and somatic cell score (SCS). A cluster of variants on chromosome 19 associated with MY, FY, and PY was identified, and another cluster on chromosome 29 associated with SCS. The most significant genomic window was located on chromosome 19, explaining up to 9.6% of the phenotypic variation for MY, 8.1% for FY, 9.1% for PY, and 1% for SCS [16].

Previous studies demonstrated that the biochemical pathways in mammary glands related to the biosynthesis and secretion of lipids, lactose, and proteins are regulated by complex gene networks [17,18]. Due to the significance of fatty acid biosynthesis, polymorphisms in the following genes were the focus of this study: butyrophilin (*BTN1A1*); acetyl-CoA carboxylase α (*ACACA*); lipoprotein lipase (*LPL*); and stearoyl-CoA desaturase (*SCD*). BTN1A1 is a milk fat globule membrane protein that plays a crucial role in lipid secretion and milk production [19]. The *BTN1A1* gene is located at chromosome 23, has eight exons and seven coding exons, respectively, and a transcription length of 3256 bps. The gene codes for 526 amino acid (AA) residues (NCBI gene 100860762). In the transcript, 219 variant alleles were described. ACACA is the primary regulatory enzyme of fatty acid biosynthesis; it catalyses acetyl-CoA conversion to malonyl-CoA [18,19]. Fatty acids are essential for forming cell membranes and are used to synthesize fat for storage in adipose tissue or secretion into milk by the mammary gland [20]. The *ACACA* gene is located at chromosome 19, consists of 52 exons, the transcription length is 6990 bps, and codes for 2329 AA residues (NCBI gene 100861224). In all, 3555 variant alleles were found in the transcript.

The *SCD* gene, which is located on goat chromosome 26, has an important role in the cellular biosynthesis of monounsaturated fatty acids (MUFAs), because most of the conjugated linoleic acids are synthesised in the mammary gland by the action of SCD in circularizing vaccenic acid [21,22]. The *SCD* mRNA has been identified by Bernhard et al., 2001 [23] and Yahyaoui et al., 2003 [24]. There are many SNPs that have been described and identified in exon 3, intron 3, intron 4, exon 5 and 6, and a deletion of a nucleotide triplet in the 3′UTR [23,24,25,26]. The *LPL* gene is involved in the hydrolysation of triglycerides to glycerol and free fatty acids and in lipoprotein transportation [27,28]. It is synthesised in the mammary gland’s epithelial cells and influences the release of fatty acids in the mammary tissue [29]. LPL enzymes are encoded by the *LPL* gene, which consists of nine exons and eight introns, for a total of 3555 nucleotides. This gene encodes a protein containing 478 amino acids, JQ670882. A few SNPs have been described in goats: a missense mutation responsible for a Ser17Thr amino acid substitution at position 17 of the signal peptide (DQ370053:c.G50C), a C2094T (DQ370053) substitution in the 3′UTR of the gene, and a substitution ss522928251:C > T in intron 7 [21,30].

Our aim was to perform an association analysis of SNPs in the *ACACA*, *BTN1A1*, *LPL*, and *SCD* genes with milk production traits: daily milk yield (DMY), protein, and fat percentage (PP, FP), and SCC of goats on organic farms in the Czech Republic.

## 2. Materials and Methods

### 2.1. Ethical Approval

The experiment was carried out under Directive 2010/63/EU of the European Parliament and European Council of 22 September 2010, on the protection of animals used for scientific purposes.

### 2.2. Animals

In this study, a total of 590 animals were included. All individuals belonged to the two Czech national goat breeds: White Shorthaired (WSH) and Brown Shorthaired (BSH) goats or their crosses with other breeds. Thus, three breed groups were determined: purebred WSH, purebred BSH, and crossbreds of both breeds, where the proportion of WSH or BSH was 50% or higher. The experiment was performed on two farms that were located in the Czech Republic. Both farms specialize in organic goat milk production, which means that goats grazed and were fed only organic feed; no hormones, antibiotics, or similar substances were applied (except a form of veterinary treatment for individuals), no genetically modified organisms were included, and animals were kept under welfare conditions according to legislation in the European Commission Regulation 889/2008, and the European Commission Regulation 834/2007. The winter feed ration consisted of haylage at approximately 2 kg a day, hay ad libitum, and a grain mix, which was dosed during milking in the milking parlour with a total amount of 300 g a day. The summer feed consisted of grass at approximately 2 kg a day (loaded in the stable), hay ad libitum, and grain mix at 300 g a day. Goats were machine-milked twice a day.

### 2.3. Performance-Testing Database

Phenotype data of genotyped animals were obtained from the performance-testing database of the Czech-Moravian Breeders’ Association. In this study, the daily milk yield (DMY) in kg, milk protein (PP) and fat (FP) per cent, and somatic cell count (SCC) were considered. The analysis of DMY, PP, and FP was performed in the group of 590 goats belonging to the White Shorthair breed (n = 490), Brown Shorthair breed (n = 76), and crossbreeds between WSH, BSH, and Saanen (n = 24). Purebred individuals were approximately 75–100% pure. The goats were sired by 37 bucks; the average number of daughters per buck was 8 (minimum 1, maximum 35). They were on the first to eleventh lactation. In all, 8640 milking records from two farms were analysed. At farm A, 2241 repeated-milk records from 279 dairy goats in 2013, 2014, and 2016 were collected. Milk records from farm B included 311 dairy goats with 6399 milk records between 2010 and 2017. Milk samples were collected repeatedly during the milking seasons. There were 2 to 49 repeated records per goat, on average, there were 14 records (approximately four milk controls per year). Milk samples for DMY, PP and FP analysis were collected throughout the whole year as follows: January (n = 571), February (n = 533), March (n = 735), April (n = 1061), May (n = 1161), June (n = 1203), July (n = 1035), August (n = 1069), September (n = 950), October (n = 74), November (n = 57), and December (n = 191).

Samples from only one farm were analysed for somatic cell count, n = 146 goats, that is, White Shorthair goats n = 100, Brown Shorthair goats n = 38, and crossbreeds n = 8. The goats were sired by 27 bucks, and the average number of daughters was 5 (minimum 1, maximum 19). Milk samples were taken during the third to eleventh lactation. The number of repeated records for SCC was 857. Data were collected during 2016 (n = 728) and 2017 (n = 129). The analysis was conducted repeatedly throughout lactation. There were 2–11 repeated records per goat, on average 5 samples per goat, with approximately three controls per year. Milk samples for SCC analysis were collected throughout the milking season as follows: March (n = 26), April (n = 148), May (n = 148), June (n = 128), July (n = 163), August (n = 124), and September (n = 120).

### 2.4. DNA Extraction and SNP Genotyping

Blood samples from 590 animals (5 mL of each) were collected from the jugular vein and preserved in 0.5 mM EDTA (pH 8.0). Genomic DNA was extracted from blood using GeneAll, Exgene^TM^, and a Clinic SV mini isolation kit (GeneAll Biotechnology cp., Ltd., Seoul, Korea; Bohemia Genetic Ltd., Prague, Czech Republic) according to the manufacturer’s recommendations. The SNPs analysed in our study are described in Table 1. SNPs in the *BTN1A1* and *LPL* genes were genotyped according to previously described methods [30,31,32].

Individual SNPs of the *ACACA* and *SCD* loci were detected by primer extension analysis with the SNaPshot Commercial Kit (Applied Biosystem, Foster City, CA, USA). Primers used for the PCR, extension analysis and electropherogram of the SNaPshot product along with the GeneScan^TM^ 120LIZTM size standard are given in Appendix A.

For the PCR reaction (512 bp fragment) of 3 SNPs in the 5′UTR [33], of *ACACA* locus, we used the set of primers, designed on the basis of the ovine sequence AJ292286 [34]. The PCR assay was performed in a 10 µL reaction mixture, consisting of 2 µL genomic DNA (10–100 ng), 1× PPP Master Mix (Top Bio Ltd., Prague, Czech Republic), and 0.5 µL (10 pmol/µL; 0.01 mM) of each primer (GENERI Biotech Ltd., Prague, Czech Republic), and sterile water up to volume. Thermal cycling conditions are presented in Appendix A.

A 536bp fragment of the *SCD* locus, at region exon3 and intron 3, was amplified by PCR with the following set of primers: F: 5′-TCCTAAgCTTATTCCAgCCCC-3′and R: 5′-gCCAgTCACTCAgAAgTACCC-3′, designed on the basis of the GenBank goat sequence (GenBank AH011188.2; AF422168.1) using Primer 3 software [35]. PCR assay was performed in a 20 µL reaction mixture consisting of 2 µL genomic DNA (10–100 ng), 1× PPP Master Mix (Top Bio Ltd., Prague, Czech Republic), of each primer (GENERI Biotech Ltd., Prague, Czech Republic) and sterile water up to volume. Thermal cycling conditions are presented in Appendix A.

The presence of fragments obtained in this phase was confirmed by gel-electrophoresis stained with ethidium bromide. The obtained PCR products were purified by using 1 unit of FastAP Thermosensitive Alkaline Phosphatase and Exonuclease I (Fermentas, Ltd., Prague, Czech Republic) to remove unwanted subproducts and incubated at 37 °C for 60 min, followed by 15 min at 85 °C.

The PEA assay utilises internal unlabelled primers which bind to a complementary PCR-generated template in the presence of AmpliTaq DNA Polymerase and fluorescently labelled ddNTPs. The polymerase extends the primer one nucleotide, adding a single ddNTP to its 3′ end. Primers were designed to allow size and colour discrimination between the different alleles (Appendix A) and were optimised to be used simultaneously.

The single-base extension (SBE) reaction for *ACACA* locus was performed in a reaction mixture with final volume of 5.0 µL, containing 1.5 µL of purified multiplex PCR product, 1× extension primer mixture (0.01 mM concentrations): K-ACACA (1206) = 0.5 µL, K-ACACA (1255) = 0.5 µL, K-ACACA (1322) = 0.5 µL, 1.5 µL deionized water, and 2.0 µL of SNaPshot Multiplex Ready Reaction Mix (Applied Biosystems, Foster City, CA, USA). The single-base extension (SBE) reaction for *SCD* locus in positions EX3_15G > A, IVS3+46 C > T, IVS3+55A > G, EX3_68A > G and IVS3+105A > G was performed in a reaction mixture with a final volume of 6.0 µL, containing 1.5 µL of purified multiplex PCR product, 1× extension primer mixture (0.01 mM concentrations): K-EX3_15G > A = 0.5 µL, K-EX3_68A > G = 0.5 µL, K- IVS3+46C > T = 1.0 µL, K-IVS3 + 55A > G = 0.5 µL K-IVS3 + 105A > G = 0.5 µL, 0.7 µL deionized water, and 2.3 µL of SNaPshot Multiplex Ready Reaction Mix (Applied Biosystems, Foster City, CA, USA). Thermal cycling consisted of 25 cycles of denaturation at 96 °C for 10 s, primer annealing at 50 °C for 5 s, and primer extension at 60 °C for 30 s (Biometra Thermoblock: 050-801 TGradient 96, Biometra, Goettingen, Germany).

For electrophoretic detection, 0.5 µL of purified multiplex SBE reaction was mixed with 0.5 µL of GeneScan-120 LIZ size standard (Applied Biosystems, Foster City, CA, USA) and 9.0 µL of Hi–DiTMFormamide (Applied Biosystems, Foster City, CA, USA), following denaturation step at 95 °C for 5 min and analysed by capillary electrophoresis using the Applied Biosystems^®^ 3130 Genetic Analyzer, an E5-Matrix Standard Set DS-02, a 36 cm capillary, and POP7 polymer (Applied Biosystems, Foster City, CA, USA). The results of genotyping were analysed and evaluated using GeneMapper v 3.5 software (Applied Biosystems, Foster City, CA, USA). The dye colour of the fragment was used to identify the nucleotide of interest (Appendix A).

### 2.5. Statistical Analysis

The dataset was edited, and unreliable data were removed. SCCs less than 13 and more than 9998 were removed from the analysis. To achieve an approximately normal distribution, the SCC was log-transformed into somatic cell score (SCS). The transformation was performed as follows:SCS = log2 (SCC/100) + 3
where SCC is somatic cell count, which is expressed in thousands per 1 mL of milk.

Hardy–Weinberg equilibrium was tested by the χ^2^ test. The effect of gene polymorphisms on milk performance traits and SCS was analysed using the PROC MIXED procedure of SAS with repeated measurements [36]. Tested effects were considered statistically significant at *p* < 0.05, but biological importance was also considered. The following linear model was used for all traits (DMY, PP, FP, SCS):Y_ijklmnop_ = µ + G_i_ + HY_j_ + month_k_ + lac_l_ + breed_m_ + sire_n_ + goat_o_ + e_ijklmno_ ,
where Y_ijklmno_ = DMY, FP, PP, SCS; G_i_ = fixed effect of the genotype (class effect i = 1, 2, 3); HY_j_ = combined fixed effect of herd-year (class effect j = 1, …, 11); month_k_ = fixed effect of the month of the year of milking (class effect l = 1, …, 12); lac_l_ = fixed effect of the lactation order (class effect l = 1, …, 9 for DMY, FP and PP or l =1, …, 9 for SCS); breed_m_ = fixed effect of the breed (class effect m = 1, …, 3); sire_n_ = random effect of the father of the goat; goat_o_= permanent environment of the goat (repeated measurement); and e_ijklmno_ = random residual effect.

The post hoc comparison was performed by Scheffe´s method. A Bonferroni correction for multiple comparisons was applied to all significant associations. The correction factor was derived from the number of SNPs tested. The significance threshold (*p* < 0.05 and *p* < 0.01) was divided by the number of tests. Thus, Bonferroni-corrected significance levels of 0.05/13 = 0.004 and 0.01/13 = 0.0008 were applied.

## 3. Results and Discussion

### 3.1. Descriptive Statistics and Phenotypic Correlations

Genotype and allelic frequencies and the number of animals and records included in the analysis are shown in Table 2.

The descriptive statistics of the analysed traits are shown in Table 3.

Phenotypic correlations between milk traits (DMY, PP, FP) and SCS are shown in Table 4. The relationships of DMY with PP and FP were negative, which corresponds to the well-known dilution effect reflecting the reductions in fat and protein contents as milk yield increases [37]. Among milk components and SCC, positive correlations were found. Some authors found a negative and significant phenotypic correlation between the logarithm of somatic cell count and milk yield [38]. Milk protein content consistently showed a significant positive correlation to the logarithm of SCC. Their study showed a similar correlation between SCC and milk yield, or milk protein content of dairy goats’ milk as found in dairy cows’ milk. However, they stated the impossibility of employing commonly used physiological parameters for dairy cows in evaluating the mammary health status of dairy goats. According to other results, the goat milk composition did not change when milk SCC varied among three groups from 214,000 to 1450,000 cells/mL [39]. In ovine milk, the components can be expected to vary independently of milk SCC [40].

### 3.2. Environmental Factors

Environmental factors play an essential role in milk production and mastitis occurrence, so controlling environmental factors could permit or prevent animals from expressing their genetic potential [15]. In our analysis, we considered the following fixed effects: the combined effect of the herd-year (HY), month of milking, lactation order, and breed. All of the mentioned factors were significant for daily milk yield, protein, and fat percentages. The exception was an effect of the breed on DMY, where significance was observed only in models with a few SNPs *LPL* g.103G > A, g.185G > T, g.257C > T, and g.300G > A. The combined effect of HY explained 7% of the total of 26% variability explained by all tested factors. The HY effect included farm management, milking routines, milking frequency, and feed quality. The Czech Republic is situated in the middle of Europe. The climate is mild with four seasons (spring, summer, autumn, winter). Seasonal kidding usually starts in January in the winter season. After this, goat milk production increases with the growing needs of kids. The month of milking reflects changes during time as well as the nutritional condition of pasture which is rising with the increasing temperature, and also the variation of climate and phase of lactation throughout the year. Milk yield is also negatively correlated with milk fat and protein contents, so with increasing milk production, the milk fat and protein levels decrease [41]. The fixed effect of a month of milking explained approximately 10% of a total of 26% variability. The highest milk production was observed between the lactations. This is in agreement with many authors [41,42,43]. During this period, the highest milk yield is probably caused by the physical appearance, size, and quality of the udder. These morphological characteristics are affected by the breed and genotype of goats as well [44]. Contrary, younger goats tend to have a higher milk fat content than older goats [44]. Milk components were significantly affected by breed; for daily milk yield, the impact of breed varied. Differences between breeds were also confirmed by many authors [42,44].

The somatic cell count could be affected by numerous factors, such as milking routine, stage of lactation, lactation order, breed, feed quality, and health status [3,5,45]. In our analysis of the SCS, only the HY effect was significant, but not the effect of month of milking, lactation order, and breed. HY comprises farm management, feed quality, milking frequency, milking hygiene, pasture quality throughout the year, etc. This analysis was performed on only one farm over multiple years. Thus, the significance of the HY effect probably indicates the difference between the tested years. A trend of increasing somatic cell counts was observed throughout the year (from March to September); the differences among succeeding months were significant or near the significance threshold. Bergonier et al., 2003, claimed that higher rates of mastitis occurrence are observed at the beginning of machine milking and during the first third of lactation, but mastitis is rarely observed during drying-off or parturition [46]. The stage of lactation was not included in this study because of a lack of data. Nevertheless, the mastitis incidence in goats does not vary with the lactation stage, in contrast with cows [46]. Goat milk contains naturally higher SCCs than cow milk due to the apocrine secretory process in goats [47]. There is no consensus on whether the SCC is related to milk production (MY, FP, PP). Several authors claimed that a significantly lower fat content was found in goat milk infected with *S. aureus* than in noninfected milk. Even an SCC of approximately 3,300,000 cells/mL might not be connected with mastitis or pathological differences in the goat mammary gland [5].

Analysed farms were specialized in organic goat milk production, so the environmental conditions and herd management might differ in feeding routine, health management, and welfare issues compared to conventional goat farms. There is an assumption that there should be differences in SCC between organic and conventional farming. Goats from conventional farming could be exposed to various chemical agents such as disinfections, mycotoxins, pesticides, or residues of antibiotics. These agents affect the development and activity of the microbiological profile of milk. However, many studies on small ruminants and dairy cattle did not confirm differences between organic and conventional farms in milk quality parameters [48,49].

### 3.3. Associations between SNP, Milk Production Traits, and SCS

In total, 14 SNPs were included in the association analysis. SNP g.1255A > G of the *ACACA* gene was excluded because of monomorphism. The associations between 13 SNPs, milk production traits (MY, FP, PP), and SCS are shown in Table 5, and the differences between genotypes are shown in Table 6.

Polymorphisms other than those in the *ACACA* gene were associated with fat content in milk. Seven SNPs, namely, *BTN1A1* g.599A > G; *LPL* g.185G > T, g.300G > A, and G50C, and *SCD* EX3_15G > A; EX3_68A > G, and IVS3+46 C > T, were detected as significant (Table 5). Significant differences between genotypes were found for five SNPs: *BTN1A1* g.599A > G; *LPL* G50C; and *SCD* EX3_15G > A, EX3_68A > G, and IVS3+46 C > T (Table 6). After Bonferroni correction, only SNPs in the *SCD* and *LPL* genes showed a significant association with fat content in milk. The non-significance of *ACACA* polymorphisms is surprising, as other authors found an effect of the gene, especially on the milk fat content, but not for milk yield and protein content in goat and sheep milk [33,50], which keeps the gene promising for the future research [51]. The role of LPL in fat synthesis was also stated by other authors [52].

All analysed genes showed an association with protein content. A strong association (*p* < 0.01) was found between *BTN1A1* g.599A > G and *LPL* g.185G > T. Other SNPs, such as *ACACA* g.1322T > C and g.257C > T and *SCD* IVS3+46 C > T, reached significant levels of *p* < 0.05 (Table 5). Significant differences between genotypes were found for *ACACA* g.1322T > C, *BTN1A1* g.599A > G, *LPL* 257C > T, and *SCD* IVS3+46 C > T (Table 6). After Bonferroni correction, the number of statistically significant SNPs decreased from 5 SNPs to just *LPL* g.185G > T. However, for this SNP, the differences between genotypes were not significant. Only two SNPs were found to be associated with daily milk yield, *ACACA* g.1322T > C (*p* < 0.05) and *LPL* g.300G > A (*p* < 0.01) (Table 5). The differences between genotypes showed only slightly significant differences (Table 5). For the SCS, the only association was detected with the SNP *BTN1A1* g.599A > G (*p* < 0.05). In this polymorphism, genotype *AA* was connected with the highest somatic cell score in milk, and significant differences were found between the *AG* and *GG* genotypes (*p* < 0.05). Unfortunately, this association was not significant after Bonferroni correction. No other significant associations were found for the SCS.

When comparing the effect of SNPs retained after Bonferroni correction and the differences among genotypes, only the *LPL* G50C and *SCD* EX_15G > A and EX3_68A > G polymorphisms showed a significant effect at the SNP and genotype levels. Additionally, other gene polymorphisms in goats are studied. For example, the AGPAT6 gene was found to significantly influence both the fat and protein contents and milk yield [53]. Additionally, the variability of the haplotypic sequences at the loci of the casein complex was studied [15]. The authors point out that a complete definition of the haplotypes in the casein complex in goats is difficult given the high genetic variability.

*LPL*, *ACACA*, and *SCD* ovine genes expression were found to be influenceable by diets [54]. The ability of nutrigenomic regulation of the transcription confirmed that these genes play a critical role in the regulation of lipid metabolism processes in sheep and could be associated with fatty acid profiles in milk and meat. Ovine LPL gene should also be studied due to its expression to microRNA-148a [55]. *SCD* and *BTN1A1* genes were not found to be differentially expressed when comparing two Spanish sheep breeds [56].

Such studies like this are frequent in cattle. Often, the polymorphisms of genes for milk proteins are analysed [57]. The authors found a significant influence of the *CSN1S2* gene on the milk protein content and also of the *DGAT1* gene on the milk yield. Others found a significant association of the *SCD1* bovine gene with fat and milk urea contents and of the *ACACA* gene with SCS [20]. Other analyses described the significant effect of *DGAT1* and *ACACA* polymorphisms on the milk performance indicators [58,59]. *DGAT1* and *SCD1* are favourite subjects in dairy cattle [60]. More extensive studies were done, comprising eleven loci and 25 indels [61]. The *BTN1A1* polymorphisms were studied even in water buffalo [62].

## 4. Conclusions

Concisely, some SNPs in the included genes showed association with milk traits but not with SCS. After more accurate Bonferroni correction, the significant associations of SNPs were only rare. Thus, our results support the generally accepted fact that environmental factors are more important than genetic for milk-production traits. However, along with population genetic analyses, the study of major genes in goats helps to better understand the genetic background of the milk-performance complex. This may contribute to future more effective selection in dairy goat populations.

## Figures and Tables

**Table 1 animals-11-01796-t001:** Analysed SNPs and methods used.

Gene	GeneBank Access. No.	SNPs	Region	AA	Method	Reference
Acetyl-CoA carboxylase α (*ACACA*)	AJ292286	g.1206C > T	3′UTR		PEA ^a^	[33]
	AJ292286	g.1255A > G	5′UTR		PEA	
	AJ292286	g.1322T > C	3′UTR		PEA	
Butyrophilin (*BTN1A1*)	NM001285618.1	g.599A > G	Exon4	Glu184/Lys	PCR-RFLP	[31]
Lipoprotein lipase (*LPL*)	KP261023	g.103G > A	signal peptide	Gly/Arg	PEA	[32]
	KP261023	g.185G > T	intron I		PEA	[32]
	KP261023	g.257C > T	intron I		PEA	[32]
	KP261023	g.300G > A	intron I		PEA	[32]
	DQ370053	G50C	signal peptide	Ser36/Thr	PEA	[30]
Stearoyl-coenzyme A desaturase (*SCD*)	AF422168.1	EX3_15G > A	Exon3	Val109/Met	PEA	[25]
	AF422168.1	EX3_68A > G	Exon3	Arg/Arg	PEA	[25]
	AF422168.1	IVS3+46C > T	Intron3		PEA	Present work
	AF422168.1	IVS3+55A > G	Intron3		PEA	[25]
	AF422168.1	IVS3+105A > G	Intron3	data	PEA	Present work

^a^ SNPs of loci were genotyped by using multiplex Primer Extension Analysis (Appendix A), AA—Amino Acid.

**Table 2 animals-11-01796-t002:** Genotype and allelic frequencies and the number of animals and records included in the analysis.

Gene	Gene Bank Access. No.	SNPs	Genotypes	N	Frequency	Allele	Frequency	*χ* ^2^	Nmilk	Nscc
Acetyl-CoA carboxylase α (*ACACA*)	AJ292286	g.1206C > T	CC	181	0.522	C	0.72	2.754	3675	490
			CT	82	0.401				1885	288
			TT	22	0.077	T	0.28		479	64
Acetyl-CoA carboxylase α (*ACACA*)	AJ292286	g.1322T > C	CT	19	0.067	C	0.03	0.237	410	49
			TT	266	0.933	T	0.97		5629	793
Butyrophilin (*BTN1A1*)	NM001285618.1	g.599A > G	AA	7	0.023	A	0.15	0.019	140	14
			AG	76	0.249				1553	199
			GG	222	0.728	G	0.85		4559	636
Lipoprotein lipase (*LPL*)	KP261023	g.103G > A	GG	265	0.892	G	0.94	0.360	3851	494
			GA	30	0.101				361	48
			AA	2	0.007	A	0.06		42	12
Lipoprotein lipase (*LPL*)	KP261023	g.185G > T	GG	210	0.707	G	0.85	0.365	2793	352
			GT	82	0.276				1414	202
			TT	5	0.017	T	0.15		47	0
Lipoprotein lipase (*LPL*)	KP261023	g.257C > T	TT	11	0.037	T	0.22	0.444	140	15
			CT	109	0.367				1669	241
			CC	177	0.596	C	0.78		2445	298
Lipoprotein lipase (*LPL*)	KP261023	g.300G > A	GG	219	0.737	G	0.86	0.257	3285	453
			GA	70	0.236				888	95
			AA	8	0.027	A	0.14		81	6
			CC	3	0.010	C	0.12	0.221	88	9
Lipoprotein lipase (*LPL*)	DQ370053	G50C	CG	68	0.224				1410	210
			GG	232	0.766	G	0.88		4752	631
Stearoyl-coenzyme A desaturase (*SCD*)	AF422168	EX3_15G > A	AA	182	0.591	A	0.76	0.791	3808	512
			AG	103	0.334				2049	262
			GG	23	0.075	G	0.24		524	83
Stearoyl-coenzyme A desaturase (*SCD*)	AF422168	EX3_68A > G	AA	256	0.831	A	0.91	0.051	5393	691
			AG	50	0.162				948	158
			GG	2	0.006	G	0.09		40	8
Stearoyl-coenzyme A desaturase (*SCD*)	AF422168		CC	234	0.760	C	0.86	2.823	4851	665
		IVS3+46 C > T	CT	62	0.201				1333	172
			TT	12	0.039	T	0.14		197	20
Stearoyl-coenzyme A desaturase (*SCD*)	AF422168		AA	23	0.075	A	0.15	15.899 **	524	83
		IVS3+55A > G	AG	48	0.156				899	142
			GG	237	0.769	G	0.85		4958	632
Stearoyl-coenzyme A desaturase (*SCD*)	AF422168	IVS3+105A > G	AA	25	0.081	A	0.25	0.770	546	86
			AG	107	0.347				2127	283
			GG	176	0.571	G	0.75		3708	488

N—number of animals in milk traits analysis (the number of animals for SCC analysis was smaller); Nmilk—number of samples with MDY, PP, FP performance; Nscs—number of samples with SCS; ** significant at *p* < 0.01.

**Table 3 animals-11-01796-t003:** Descriptive statistics of goat milking traits of analysed data (mean ± standard deviation).

Trait	N	Mean ± SD	Min	Max
Daily milk yield (L)	8640	2.94 ± 1.004	0.8	6.4
Milk fat percentage (%)	8640	3.06 ± 0.644	2.0	5.0
Milk protein percentage (%)	8640	3.02 ± 0.324	2.2	4.2
SCC ^1^	857	1353.52 ± 1608.240	19.0	9625.0
SCS ^2^	857	5.87 ± 1.694	0.60	9.59

^1^ SCC—somatic cell count in thousands per 1 mL of milk, ^2^ SCS = log2 (SCC/100) + 3, SCC—somatic cell count.

**Table 4 animals-11-01796-t004:** Pearson correlation coefficients between all analysed traits.

Trait	DMY	PP	FP
PP	−0.205 ** ± 0.98		
FP	−0.154 ** ± 0.66	0.401 ** ± 0.61	
SS	−0.418 ** ± 1.54	0.154 ** ± 1.67	0.113 ** ± 1.68

DMY—daily milk yield; PP—milk protein %; FP—milk fat %; SCS = somatic cell score; ** *p* < 0.01.

**Table 5 animals-11-01796-t005:** Significance of fixed effects for milk traits and somatic cell score—individual SNPs.

		Daily Milk Yield (L)								SCS		
SNPs	SNP	SNP ^BC^	Herd-Year	Month of Milking	Lactation Order	Breed	SNP	SNP ^BC^	Herd-Year	Month of Milking	Lactation Order	Breed
g.1206C > T	ns	ns	**	**	**	ns	ns	ns	**	ns	ns	ns
g.1322T > C	*	ns	**	**	**	ns	ns	ns	**	ns	ns	ns
g.599A > G	ns	ns	**	**	**	ns	*	ns	**	ns	ns	ns
g.103G > A	ns	ns	**	**	**	*	ns	ns	**	ns	ns	ns
g.185G > T	ns	ns	**	**	**	*	ns	ns	**	ns	ns	ns
g.257C > T	ns	ns	**	**	**	*	ns	ns	**	ns	ns	ns
g.300G > A	**	*	**	**	**	*	ns	ns	**	ns	ns	ns
G50C	ns	ns	**	**	**	ns	ns	ns	**	ns	ns	ns
EX3_15G > A	ns	ns	**	**	**	ns	ns	ns	**	ns	ns	ns
EX3_68A > G	ns	ns	**	**	**	ns	ns	ns	**	ns	ns	ns
IVS3+46 C > T	ns	ns	**	**	**	ns	ns	ns	**	ns	ns	ns
IVS3+55A > G	ns	ns	**	**	**	ns	ns	ns	**	ns	ns	ns
IVS3+105A > G	ns	ns	**	**	**	ns	ns	ns	**	ns	ns	ns
	Protein (%)									Fat (%)		
g.1206C > T	ns	ns	**	**	**	*	ns	ns	**	**	**	**
g.1322T > C	*	ns	**	**	**	*	ns	ns	**	**	**	**
g.599A > G	**	ns	**	**	**	*	**	ns	**	**	**	**
g.103G > A	ns	ns	**	**	**	**	ns	ns	**	**	**	**
g.185G > T	**	**	**	**	**	*	*	ns	**	**	**	**
g.257C > T	*	ns	**	**	**	*	ns	ns	**	**	*	**
g.300G > A	ns	ns	**	**	**	**	**	*	**	**	*	**
G50C	ns	ns	**	**	**	*	**	**	**	**	*	**
EX3_15G > A	ns	ns	**	**	**	*	**	*	**	**	*	**
EX3_68A > G	ns	ns	**	**	**	*	**	*	**	**	**	**
IVS3+46C > T	*	ns	**	**	**	*	**	ns	**	**	*	**
IVS3+55A > G	ns	ns	**	**	**	*	ns	ns	**	**	**	**
IVS3+105A > G	ns	ns	**	**	**	*	**	ns	**	**	*	**

SCS—somatic cell score; SNP ^BC^—*p*-value after Bonferroni correction of SNP in each row are significant as computed in model with effects of individual SNP, herd-year, month, lactation order, and breed * significant at *p* < 0.05; ** significant at *p* < 0.01; ns not significant.

**Table 6 animals-11-01796-t006:** Daily milk yield, protein, and fat percentage, and somatic cell score (LSM ± SE) and differences among genotypes.

SNPs	Genotype	DMY	PP	FP	SCS
*ACACA* g.1206C > T	CC	2.50 ± 0.112	3.16 ± 0.037	3.23 ± 0.060	6.25 ± 0.461
	CT	2.55 ± 0.110	3.14 ± 0.036	3.20 ± 0.059	6.24 ± 0.448
	TT	2.50 ± 0.139	3.15 ± 0.043	3.16 ± 0.072	7.16 ± 0.660
*ACACA* g.1322T > C	CT	2.75 ± 0.135 ^a^	3.09 ± 0.043 ^a^	3.19 ± 0.071	6.04 ± 0.725
	TT	2.52 ± 0.105 ^a^	3.15 ± 0.035 ^a^	3.21 ± 0.058	6.36 ± 0.413
*BTN1A1* g.599A > G	AA	2.74 ± 0.177	3.13 ± 0.051	3.09 ± 0.088	7.07 ± 1.087
	AG	2.51 ± 0.108	3.17 ± 0.034 ^A^	3.24 ± 0.057 ^a^	7.04 ± 0.477 ^a^
	GG	2.53 ± 0.106	3.12 ± 0.034 ^A^	3.17 ± 0.056 ^a^	6.08 ± 0.418 ^a^
*LPL* g.103G > A	GG	2.70 ± 0.119	3.13 ± 0.045	3.18 ± 0.071 ^A^	6.21 ± 0.588
	GA	3.07 ± 0.162	3.10 ± 0.056	3.33 ± 0.093 ^A,B^	5.92 ± 0.837
	AA	3.07 ± 0.267	3.18 ± 0.082	2.90 ± 0.142 ^B^	5.99 ± 1.378
*LPL* g.185G > T	GG	2.67 ± 0.115	3.11 ± 0.047	3.14 ± 0.077	6.23 ± 0.622
	GT	2.72 ± 0.113	3.14 ± 0.045	3.18 ± 0.074	6.10 ± 0.632
	TT	Not est.	Not est.	Not est.	Not est.
*LPL* g.257C > T	TT	2.81 ± 177	3.27 ± 0.062 ^A,B^	3.18 ± 0.107	6.67 ± 1.179
	CT	2.69 ± 0.112	3.15 ± 0.046 ^A^	3.20 ± 0.077 ^A^	6.07 ± 0.601
	CC	2.69 ± 0.117	3.11 ± 0.047 ^B^	3.10 ± 0.079 ^A^	6.26 ± 0.658
*LPL* g.300G > A	GG	2.69 ± 0.111	3.13 ± 0.044	3.16 ± 0.074	6.22 ± 0.596
	GA	2.70 ± 0.124	3.15 ± 0.048	3.21 ± 0.081	5.94 ± 0.699
	AA	2.80 ± 0.300	3.32 ± 0.091	3.08 ± 0.163	7.02 ± 1.854
*LPL* G50C	CC	2.61 ± 0.233	3.05 ± 0.066	2.70 ± 0.114 ^A,B^	6.24 ± 1.324
	CG	2.59 ± 0.116	3.13 ± 0.037	3.22 ± 0.061 ^B^	6.53 ± 0.508
	GG	2.51 ± 0.105	3.15 ± 0.034	3.19 ± 0.06 ^A^	6.33 ± 0.418
*SCD* EX3_15G > A	AA	2.52 ± 0.105	3.15 ± 0.034	3.21 ± 0.057 ^A^	6.33 ± 0.415
	AG	2.53 ± 0.110	3.13 ± 0.035	3.14 ± 0.059 ^A^	6.30 ± 0.494
	GG	2.38±0.130	3.16 ± 0.040	3.24 ± 0.068	7.00 ± 0.679
*SCD* EX3_68A > G	AA	2.51 ± 0.104	3.15 ± 0.034	3.21 ± 0.056 ^A^	6.36 ± 0.414
	AG	2.53 ± 0.120	3.13 ± 0.038	3.14 ± 0.062 ^B^	6.36 ± 0.592
	GG	2.18 ± 0.277	3.20 ± 0.078	3.55 ± 0.136 ^A,B^	6.62 ± 1.738
*SCD* IVS3+46C > T	CC	2.52 ± 103	3.15 ± 0.034 ^A^	3.21 ± 0.056 ^A^	6.33 ± 0.412
	CT	2.52 ± 0.112	3.15 ± 0.038 ^B^	3.15 ± 0.060 ^A^	6.59 ± 0.519
	TT	2.43 ± 0.157	3.05 ± 0.047 ^A,B^	3.15 ± 0.081	7.00 ± 0.916
*SCD* IVS3+55A > G	AA	2.35 ± 0.134	3.17 ± 0.040	3.26 ± 0.067	6.92 ± 0.693
	AG	2.50 ± 0.125	3.14 ± 0.038	3.18 ± 0.064	6.10±0.609
	GG	2.52 ± 0.107	3.15 ± 0.034	3.20 ± 0.056	6.34 ± 0.410
*SCD* IVS3+105A > G	AA	2.33 ± 0.128	3.16 ± 0.040	3.27 ± 0.067 ^A^	6.64 ± 0.665
	AG	2.51 ± 0.110	3.14 ± 0.035	3.16 ± 0.059 ^A^	6.16 ± 0.493
	GG	2.54 ± 0.105	3.15 ± 0.034	3.21 ± 0.057	6.38 ± 0.416

DMY—daily milk yield, PP—milk protein percentage, FP—fat protein percentage, SCS—somatic cell score. ^a,b^ Differences between genotypes with the same letters in the same column are significant at *p* < 0.05. ^A,B^ Differences between genotypes with the same letters in the same column are significant at *p* < 0.01.

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
