# Peer review of "A Candidate Gene Association Study for Economically Important Traits in Czech Dairy Goat Breeds"

_animals, 2021, doi:10.3390/ani11061796_

Round 1
Reviewer 1 Report
Authors carried out an SNP study on four candidate genes, meaning 14 SNPs to reveal the association of gene polymorphism and milk production traits of goat.
Comments to authors:
Title: please mention the species in the title
Introduction: there is no enough information on BTN1A1 and ACACA genes (compared to the others used in the study)
2.2. Animals: please explain how breed was used in the statistical analyses as manuscript says "most individuals belong to one of to two Czech goat breeds"
L184-188: provide more details on the method
L190: PEA instead of PAE
Table 2: meaning of abbrev. AA is missing
Section 3.3.: there is no discussion and comparison of literature data and the results of present study, some relevant infromation of cow studies could be useful
Author Response
Dear reviewer, thank you for your comments. I am sending you our answers and comments.
Reviewer no.1
Q: Title: please mention the species in the title
A: Accepted.
Q: Introduction: there is no enough information on BTN1A1 and ACACA genes (compared to the others used in the study).
A: The additional information was set to the Introduction section. See lines 83-86 and 88-90.
Q: 2.2. Animals: please explain how breed was used in the statistical analyses as manuscript says "most individuals belong to one of to two Czech goat breeds“.
A: The sentence was rewritten to be more clear. All individuals belong to the two Czech national goat breeds: White Shorthaired (WSH) and Brown Shorthaired (BSH) goats or their crosses with other breeds. Thus, three breed groups were determined: purebred WSH, purebred BSH, crossbreds of WSH and BSH where the proportion of WSH or BSH was 50 % or higher. See lines 113-116.
Q: L184-188: provide more details on the method.
A: Accepted. See lines 156-197.
Q: L190: PEA instead of PAE.
A: Accepted.
Q: Table 2: meaning of abbrev. AA is missing.
A: Abbreviation AA was added to Table. Table 2 was renamed as Table 1.
Q: Section 3.3.: there is no discussion and comparison of literature data and the results of present study, some relevant information of cow studies could be useful.
A: Discussion was added. See lines 318-332.
Reviewer 2 Report
Dear Authors,
Brzáková et al. analysed 14 SNP, in four different genes, previously associated with milk production traits and the somatic cell count. The authors verified the influence of these SNP in two Czech 16 dairy goat breeds, White Shorthaired and Brown Shorthaired goats. Although these two breeds are native to Czech Republic and thus more significant for this country, the results of the study are relevant for improving the production of goat's milk.
My suggestions and considerations are following and in the version of the manuscript that I have attached in the system.
Line 48 to 50: Many authors or many studies? Please insert references here…
Line 64- 67: Here, you are talking about environment factors, than is necessary to clarify where the study was performed [13].
Line 81: study instead of paper
Line 82: acetyl-CoA carboxylase α, please correct the name of the gene here and throughout the manuscript.
Line 88: The SCD gene,
Please, please write all gene names in italics throughout the manuscript.
Line 91: mRNA
Line 91-97: rewrite this sentence to make it clear that you wanted to talk about the same gene.
In addition, please, write correctly the sequence variant nomenclature (see http://varnomen.hgvs.org/) here and throughout the manuscript.
Line 98: The LPL gene
Line 102: 3,555
Line 103-104: write correctly the sequence variant nomenclature
Materials and Methods
Line 115-116: Make clear the exact number of animals of each breed used in the study.
Line 117-124: delete or transfer this information to another place in the text.
Line 42: four
Line 126: about the statement “Both farms specialize in organic goat milk production.” Better characterization of the farm's organic production system is necessary in the M&M section. In addition, the discussion section included a consideration of how organic production system influenced its results, especially those related to SCC.
Line 137-161: Rewrite. Explain the sampling better, perhaps a table is more appropriate and will facilitate the understanding of the reader. .
Line 162-172 and Table 1: these are results and should be to transfer to results section.
Line 172-175: change to 2.5. Statistical Analysis
Line 179: “TM” superscript symbol here and throughout the manuscript.
Line 190: the PAE method previously described [32].
Line 190-191 and 195-196: if you used to primer previously described [32], please delete this sentence “The primers for PCR amplification were designed using Primer 3 software (http://bioinfo.ut.ee/primer3-0.4.0/primer3; AJ292286).” and “Thermal cycling conditions for ACACA included an initial denaturation step at 95°C for 2 min, followed by 35 cycles of 95°C for 30 s, 63.9°C for 45 s, and 72°C for 1 min, with a final extension step at 72°C for 5 min.” On the contrary, explain what were the modifications of the methodology described by [32].
Line 194: write primer concentration in “mM”.
Table 3: please see my suggestions in the attached manuscript file.
Table 3: why was the SSD (SCC¹) so high? To explain it…
Line 267 [3.3. Association between SNP, milk production traits and SCS] In this section, please, write correctly the sequence variant nomenclature (see http://varnomen.hgvs.org/)
Discussion/Conclusion:
The authors report the following statement at the conclusion of the abstract: “The importance of the herd-year effect on all milk performance indicators was confirmed, as was the month of milking and lactation order.” Perhaps for a Czech reader, it is not, however, for the rest of the readers the discussion of the statement shown in lines 236-259 is superficial. Therefore, it is necessary to better characterize the seasonal differences (within a year) that would reflect on the environmental conditions and on the management of the animals to the point of interfering in the phenotype controlled by these SNP.
Table S1 Legend: Please enter the name of the gene in full.

Author Response
Dear reviewer, thank you for your comments. I am sending you our answers and comments.
Reviewer no.2
Q: Line 48 to 50: Many authors or many studies? Please insert references here…
A: Accepted. Line 50.
Q: Line 64- 67: Here, you are talking about environment factors, than is necessary to clarify where the study was performed [13].
A: Accepted, line 64.
Q: Line 81: study instead of paper
A: Accepted. Line 81.
Q: Line 82: acetyl-CoA carboxylase α, please correct the name of the gene here and throughout the manuscript.
A: Accepted.
Q: Line 88: The SCD gene
A: Accepted. Line 91.
Q: Please, please write all gene names in italics throughout the manuscript.
A: Accepted.
Q: Line 91: mRNA
A: Accepted, line 94.
Q: Line 91-97: rewrite this sentence to make it clear that you wanted to talk about the same gene.
Accepted, lines 94-96.
In addition, please, write correctly the sequence variant nomenclature (see http://varnomen.hgvs.org/) here and throughout the manuscript.
Q: Line 98: The LPL gene
A: Accepted. Line 97.
Q: Line 102: 3,555
A: Accepted. Line 100.
Q: Line 103-104: write correctly the sequence variant nomenclature
A: Accepted, lines 101-103.
Materials and Methods
Q: Line 115-116: Make clear the exact number of animals of each breed used in the study.
A: Exact numbers are shown in chapter 2.3 – Performance testing database. See lines 128-131 and 139-141.
Q: Line 117-124: delete or transfer this information to another place in the text.
A: The text was deleted.
Q: Line 42: four
A: Accepted. Line 135.
Q: Line 126: about the statement “Both farms specialize in organic goat milk production.” Better characterization of the farm's organic production system is necessary in the M&M section. In addition, the discussion section included a consideration of how organic production system influenced its results, especially those related to SCC.
A: The statement about organic farms was added to the M & M section (lines 116-120) and discussion (lines 286-292).
Q: Line 137-161: Rewrite. Explain the sampling better, perhaps a table is more appropriate and will facilitate the understanding of the reader.
A: Accepted. The text was rewritten (see lines 126-147).
Q: Line 162-172 and Table 1: these are results and should be to transfer to results section.
A: Accepted. Lines 205-208 and Table 2 (line 231).
Q: Line 172-175: change to 2.5. Statistical Analysis
A: Accepted.
Q: Line 179: “TM” superscript symbol here and throughout the manuscript.
A: Accepted.
Q: Line 190: the PAE method previously described [32].
A: The chapter DNA extraction and SNP genotyping was rewritten. Linea 156-197.
Q: Line 190-191 and 195-196: if you used to primer previously described [32], please delete this sentence “The primers for PCR amplification were designed using Primer 3 software (http://bioinfo.ut.ee/primer3-0.4.0/primer3; AJ292286).” and “Thermal cycling conditions for ACACA included an initial denaturation step at 95°C for 2 min, followed by 35 cycles of 95°C for 30 s, 63.9°C for 45 s, and 72°C for 1 min, with a final extension step at 72°C for 5 min.” On the contrary, explain what were the modifications of the methodology described by [32].
A: Accepted. Chapter 2.4. DNA Extraction and SNP Genotyping.
Q: Line 194: write primer concentration in “mM”.
A: Accepted.
Q: Table 3: why was the SSD (SCC¹) so high? To explain it…
A: SCC is expressed like the number of pathogens in milk. It ranges from 19 to 9,625. SCS is expressed like log transformation of SCC to ensure normal distribution of the trait. The range of SCS is between 0.60 to 9.59. This is the reason for differences in SSD between traits.
Q: Line 267 [3.3. Association between SNP, milk production traits and SCS] In this section, please, write correctly the sequence variant nomenclature (see http://varnomen.hgvs.org/)
A: We are not sure which nomenclature is the problem. The sequence variant nomenclature was written according to Zhang et al. 2010 and Crepaldi et al. 2013. Please could you provide us more information about our failure?
Discussion/Conclusion:
Q: The authors report the following statement at the conclusion of the abstract: “The importance of the herd-year effect on all milk performance indicators was confirmed, as was the month of milking and lactation order.” Perhaps for a Czech reader, it is not, however, for the rest of the readers the discussion of the statement shown in lines 236-259 is superficial. Therefore, it is necessary to better characterize the seasonal differences (within a year) that would reflect on the environmental conditions and on the management of the animals to the point of interfering in the phenotype controlled by these SNP.
A: Accepted. See chapter 3.2. Environmental factors.
Table S1 Legend: Please enter the name of the gene in full.
A: Accepted.
Reviewer 3 Report
The manuscript “A candidate gene association study for economically important traits in Czech dairy breeds” analyzed the association of SNPs in four different genes with several milk production traits in dairy goats. The sample size is enough, and the experimental design is reasonable. However, I have some concerns about the statistical methods as some fixed effects have not been well described, and the haplotype analyses were not available. The manuscript requires some English editing as well.
Comments
I suggest the authors add dairy goats in the title.
Line 15: Might change “animal production” to “animal production in the country” as you should not generalize it for all the countries.
Line 16: Define SNPs
Line 16: Change 4 to four
Line 19: remove the before somatic cell count.
Line 20: Remove this one, “ In particular, the milk quality and health status of the mammary gland should be considered,” or rewrite it as it is not well connected to the previous text.
Line 22 and 23: might change “including genetic and environmental factors” to “including genetic, environmental factors and their interactions.”
Line 26: Define SNPs, PCR, and PCR-RFLP before using them in the abstract.
Line 28-29: Are all SNPs significantly associated? Did the authors mean the genes or SNPs in the genes?
Line 29: How did the authors test the association of the genes with traits?
Line 30: What the threshold for Bonferroni correction and the significance threshold?
Line 33: it is not clear what the authors mean “as was the month of milking and lactation order”. The authors might add a sentence for a conclusion to make the abstract more complete.
Line 45: might remove the in the somatic cell count.
Line 47: Which milk products the authors mean here.
Line 49: “The heritabilities of milk yield have been found to be low to moderate, similar to cows.” Might change as “ Like cows, the heritabilities of milk yield in goats have been found to be low to moderate.”
Line 50: Might remove “ongoing” in the sentence.
Line 51: Is the somatic cell count the same as the somatic cell score here?
Line 50-59: It is better to add the reference in each sentence or in the first sentence the authors described the study. Also, the authors should add the population size of their studies.
Line 62; Change somatic cell score to SCS.
Line 70-71: Which is the study?
Line 77; The sentence is not quite complete; which type of genomic approach and applied to what?
Line 83-84; Add the reference for this statement of BTN1A1.
Line 108: somatic cell count change to SCC
Line 115-116: Are most or all animals?
Line 134-135: The authors should make the abbreviation of traits at the first appearance.
Table 3: Are the trait named milk fat percentage and milk protein percentage?
What did not the authors estimate the genetic correlations?
If the authors did the PCR-RFLP, please provide more detail about the enzyme and step to identify the bands in the manuscript (not just the supplementary file)
Line 210: Are the milk fat percentage and milk protein percentage recorded daily or for each lactation cycle?
The authors could you the abbreviation for the traits here.
Why did the authors not include the farm effects in the models?
Line 218-219: How did the authors identify the haplotypes; how many haplotypes did the authors have, how did the authors get 16 classifications of haplotypes?
Table 5 is not clear and not necessary to ad the p-value as it; the authors just simply mark the effects which are significant.
There are two columns named Gene
Line 214: why did the authors have a class of lactation from 1 to 11 as the authors collected the milk from the third lactation?
For the breed, change l to m/
Line 230-231: In which species? No relationships or no significant correlations? Please give details for discussion.
Table 4: How did the authors identify the correlations? What are the SEs of the correlations?
Line 239: What did the authors mean by lactation order?
Line 240-242: It is not clear; please break the sentence into two sentences.
Line 240-242: What is the amount? Give the range.
Line 309 -310; It is not so clear. Is SCS a milk trait?
Line 309-312: The authors do not need to repeat the results,
Line 313: What is the mean of crucial here.
What did the authors mean by “Constituents”?
Line 316-317: What did the authors mean here? What did the authors mean by “the environment and breeding conditions”. What are human research or other non-breeding studies?
Author Response
Dear reviewer, thank you for your comments. I am sending you our answers and comments.
Reviewer no.3
Q: I suggest the authors add dairy goats in the title.
A: Accepted.
Q: Line 15: Might change “animal production” to “animal production in the country” as you should not generalize it for all the countries.
A: Accepted. Line 17.
Q: Line 16: Define SNPs
A: Accepted. Line 18.
Q: Line 16: Change 4 to four
A: Accepted. Line 18.
Q: Line 19: remove the before somatic cell count.
A: Accepted.
Q: Line 20: Remove this one, “ In particular, the milk quality and health status of the mammary gland should be considered,” or rewrite it as it is not well connected to the previous text.
A: The text was removed, lines 36-37.
Q: Line 22 and 23: might change “including genetic and environmental factors” to “including genetic, environmental factors and their interactions.”
A: Accepted. Line 25.
Q: Line 26: Define SNPs, PCR, and PCR-RFLP before using them in the abstract.
A: SNPs were defined in summary. However, SNP, PCR and PCR-RFLP are generally used abbreviations in this area so we do not consider it important to explain these abbreviations.
Q: Line 28-29: Are all SNPs significantly associated? Did the authors mean the genes or SNPs in the genes?
A: All tested genes were significantly associated with milk traits, but not with all SNPs. The following sentences were rewritten to be more clear. See lines 31-33.
Q: Line 29: How did the authors test the association of the genes with traits?
A: The information was added. Line 30-31.
Q: Line 30: What the threshold for Bonferroni correction and the significance threshold?
A: Bonferroni corrected significance levels were 0.05/13 = 0.004 and 0.01/13 = 0.0008. See chapter 2.5. Statistical Analysis.
Q: Line 33: it is not clear what the authors mean “as was the month of milking and lactation order”. The authors might add a sentence for a conclusion to make the abstract more complete.
A: The sentence was rewritten. It meant the importance of environmental factors (herd-year effect, effect of month when the milk samples were collected and effect of lactation order). See line 36-37.
Q: Line 45: might remove the in the somatic cell count.
A: Accepted.
Q: Line 47: Which milk products the authors mean here.
A: The sentence was rewritten. Line 48-49.
Q: Line 49: “The heritabilities of milk yield have been found to be low to moderate, similar to cows.” Might change as “ Like cows, the heritabilities of milk yield in goats have been found to be low to moderate.“
A: Accepted. See line 50-51.
Q: Line 50: Might remove “ongoing” in the sentence.
A: Accepted.
Q: Line 51: Is the somatic cell count the same as the somatic cell score here?
A: No, somatic cell count (SCC) is used to monitor udder health and it represents the number of pathogens in milk. Somatic cell score is a log transformation of SCC. This log transformation was used for analysis because it achieves a nearly normal distribution in contrast with SCC.
Q: Line 50-59: It is better to add the reference in each sentence or in the first sentence the authors described the study. Also, the authors should add the population size of their studies.
A: Accepted. References were added. See lines 50-62.
Q: Line 62; Change somatic cell score to SCS.
A: Accepted. Line 63.
Q: Line 70-71: Which is the study?
A: Reference was added. Line 70.
Q: Line 77; The sentence is not quite complete; which type of genomic approach and applied to what?
A: It was a genome-wide association study. The text was completed. Lines 73-78.
Q: Line 83-84; Add the reference for this statement of BTN1A1.
A: The reference was added – Qu et al. 2011. Line 83.
Q: Line 108: somatic cell count change to SCC
A: Accepted, line 105.
Q: Line 115-116: Are most or all animals?
A: All animals. The part was rewritten to be more clear. Line 113.
Q: Line 134-135: The authors should make the abbreviation of traits at the first appearance.
A: Done, the abbreviations are first used at the end of Introduction. Line 105.
Q: Table 3: Are the traits named milk fat percentage and milk protein percentage?
A: Yes, it was accepted and corrected.
Q: What did not the authors estimate the genetic correlations?
A: Genetic correlation between traits were not estimated because of insufficient number of records in this study.
Q: If the authors did the PCR-RFLP, please provide more detail about the enzyme and step to identify the bands in the manuscript (not just the supplementary file)
A: Accepted. The chapter 2.4. was rewritten to be more complete.
Q: Line 210: Are the milk fat percentage and milk protein percentage recorded daily or for each lactation cycle?
A: No, milk performance (milk fat percentage and milk protein percentage) were collected randomly throughout lactation. On average, 4 milk samples were collected per lactation. On average 14 milk controls were performed per goat throughout all analysed lactations. See lines 132-135 and 142-144.
Q: The authors could use the abbreviation for the traits here.
A: Accepted. Line 215.
Q: Why did the authors not include the farm effects in the models?
A: The farm effect was included in the combined fixed effect of herd and year (HY).
Q: Line 218-219: How did the authors identify the haplotypes; how many haplotypes did the authors have, how did the authors get 16 classifications of haplotypes?
A: It was our fault, haplotypes were finally not included in this study. All mentions about haplotypes have been deleted.
Q: Table 5 is not clear and not necessary to add the p-value as it; the authors just simply mark the effects which are significant. There are two columns named Gene
A: Accepted. Table was edited.
Q: Line 214: why did the authors have a class of lactation from 1 to 11 as the authors collected the milk from the third lactation?
A: Corrected, in Material and methodology. Line 217.
Q: For the breed, change l to m/
A: Accepted. Line 218.
Q: Line 230-231: In which species? No relationships or no significant correlations? Please give details for discussion.
A: The discussion was extended and written more clearly. See 239-245.
Q: Table 4: How did the authors identify the correlations? What are the SEs of the correlations?
A: The correlation was calculated by software SAS using the PROC CORR procedure. It is phenotype correlation, genetic correlations were not calculated. SEs of the correlations were calculated by software STATISTICA. Standard errors were added to table 4.
Q: Line 239: What did the authors mean by lactation order?
A: It is the number of lactations - for example first, second or third lactation.
Q: Line 240-242: It is not clear; please break the sentence into two sentences.
A: Accepted. See 251-255.
Q: Line 240-242: What is the amount? Give the range.
A: The combined effect of HY explained approximately 7 % of variability from a total of 26 % explained by the all tested factors. Lines 254 and 261.
Q: Line 309 -310; It is not so clear. Is SCS a milk trait?
A: Corrected, conclusion was rewritten. See lines 346-351.
Q: Line 309-312: The authors do not need to repeat the results
A: The section Conclusion was rewritten. Lines 346-351.
Q: Line 313: What is the mean of crucial here.
A: It means an important effect. These effects explained a lot of total trait variability. The section Conclusion was rewritten. Lines 346-351.
Q: What did the authors mean by “Constituents”?
A: The section Conclusion was rewritten. Lines 346-351.
Q: Line 316-317: What did the authors mean here? What did the authors mean by “the environment and breeding conditions”. What are human research or other non-breeding studies?
A: The section Conclusion was rewritten. Lines 346-351.
Round 2
Reviewer 2 Report
Please, see attached file

Author Response
Dear reviewer, thank you for your comments. All your requirements were accepted.
Reviewer 3 Report
Dear Authors,
Thank you for addressing my comments.
The manuscript has been significantly improved. I have some other comments:
Like 50-52: Insert citation to support the sentence.
Line 76-77: If the authors mentioned the same study as the above sentences, then please insert reference [16] into the sentence.
Line 90-107. The authors can move into one to two paragraphs as the current form of the paragraphs are too short (one or two sentences).
The same comment for lines 125 to 146, 172-190, lines 309-321.
Table 4: Please remove “(Figures in parenthesis indicate the number of observations)” in the title as well as remove the number of samples within the table.
Line 322: Change diet to diets
Line 348: Might clarify “the factors”
Line 347-351: it is the results, not the conclusion. Please remove them.
Non-genetic factors more important than what? Did the authors test the genetic factors? It is quite obvious that environmental factors important for all the milk production traits.
Author Response
Dear reviewer, thank you for your comments. I am sending you our answers and comments.
Q: Like 50-52: Insert citation to support the sentence.
A: Accepted. Line 51.
Q: Line 76-77: If the authors mentioned the same study as the above sentences, then please insert reference [16] into the sentence.
A: Accepted. Line 77.
Q: Line 90-107. The authors can move into one to two paragraphs as the current form of the paragraphs are too short (one or two sentences). The same comment for lines 125 to 146, 172-190, lines 309-321.
A: Accepted.
Q: Table 4: Please remove “(Figures in parenthesis indicate the number of observations)” in the title as well as remove the number of samples within the table.
A: Accepted.
Q: Line 322: Change diet to diets
A: Accepted.
Q: Line 348: Might clarify “the factors”
A: Accepted.
Q: Line 347-351: it is the results, not the conclusion. Please remove them.
Non-genetic factors more important than what? Did the authors test the genetic factors? It is quite obvious that environmental factors important for all the milk production traits.
A: The conclusion was rewritten.